# Genetic and Ecological Approaches to Introduced Populations of Pumpkinseed Sunfish (*Lepomis gibbosus*) in Southwestern Europe

**Angela Lambea-Camblor** [1], **Felipe Morcillo** [2], **Jesús Muñoz** [3] and **Anabel Perdices** [1,*]

1   Department of Biodiversity and Evolutionary Biology, Museo Nacional de Ciencias Naturales (MNCN-CSIC),
    C/José Gutiérrez Abascal, 2, 28006 Madrid, Spain; angela.lambea@gmail.com
2   Department of Biodiversity, Ecology and Evolution, Universidad Complutense de Madrid,
    C/José Antonio Novais, 12, 28040 Madrid, Spain; fmorcill@ucm.es
3   Real Jardín Botánico (RJB-CSIC), Pl. Murillo, 2, 28014 Madrid, Spain; jmunoz@rjb.csic.es
*   Correspondence: aperdices@mncn.csic.es

**Abstract:** Freshwater systems are among the most affected by the introduction of exotic species. The pumpkinseed sunfish *Lepomis gibbosus*, a centrarchid native to eastern North America, is listed among the top ten introduced freshwater fishes with the greatest ecological impact globally. Despite this, genetic and evolutionary studies of the species are still scarce. Here, we analyzed the genetic variability of introduced populations of *L. gibbosus* using three mitochondrial genes (COI, d-loop, and ND1). In addition, we used species distribution modeling to compare the niche of introduced versus native populations to assess the present and potential future distribution of the species under different climate change scenarios. Compared with the native populations, introduced ones present a lower level of genetic variability, indicating these populations originated from a small number of individuals from the native (Atlantic) population in the USA and Canada. The low variability was likely driven by a founder effect and subsequent bottleneck, as often occurs in invasive species. Our modeling results suggest not only that *L. gibbosus* modified its niche during the invasion process in Europe but also the possible global expansion of the species under future climatic conditions, which could facilitate its establishment in almost all continents.

**Keywords:** freshwater fishes; invasive species; mitochondrial DNA; niche overlap; species distribution models

## 1. Introduction

Biological invasions are one of the main causes of biodiversity loss on a global scale, with around 60% of current extinctions related to the presence of non-native species, also known as alien species [1]. Species introductions and translocations outside their natural distribution are a direct consequence of globalization and human activities. In recent decades, the presence of alien species in Europe, which has increased exponentially, is often triggered by changes in climatic regimes that allow the establishment of tropical species in temperate areas and the translocation of species with wide ecological plasticity between areas of similar climates [2], but also by shifts in niche width during the invasion process [3]. Indeed, several ecological studies have also shown the important role that niche plays in the invasion process [4–6].

Species distribution models (SDMs) are a powerful tool for understanding the establishment of alien species in new areas. They can be used to study niche changes during the invasion process, which can result from a species being released from some biotic and abiotic constraints that exist in its native area or adaptations to the new range that allows it to expand into new niches [3]. Species distribution models estimate the theoretical niche of a species by relating occurrence data with a set of variables that may affect its distribution and projecting it onto any geographic space [7,8].

Freshwater systems are among the most affected by alien species worldwide [9]. These ecosystems are particularly vulnerable to introductions because they not only have a greater dispersal potential than terrestrial systems [9] but also are one of the ecosystems most altered by humans [10]. Within these ecosystems, freshwater fishes are a group of great concern: they have one of the highest rates of endemicity but also the highest number of invasive species [11,12]. Globally, up to 57% of endemic fishes are threatened by invasive species [13]. In the Mediterranean, where endemic freshwater fishes are among the most threatened biota in the world [14], these percentages are much higher. For example, in the Iberian freshwater systems, more than 80% of fish species are endemic [15], and more than 90% of them are threatened to some degree, according to the IUCN [16].

Of the hundreds of introduced or invasive freshwater fish species found worldwide [17], one of the best established is *Lepomis gibbosus* (Linnaeus, 1758), also known as pumpkinseed sunfish [18]. This centrarchid is native to lakes and rivers in eastern North America [19] but has been intentionally introduced to other regions of North America and other continents for sport fishing, aquaculture, and ornamental fish trade [20,21]. The pumpkinseed was first recorded in Europe, specifically in France and Germany, in the early 1880s [22–24]. A few decades later (1910–1913), it was recorded in Bañolas Lake in northeastern Spain [25]. European populations of *L. gibbosus* are known to adapt quickly to high temperatures [26] and human-modified areas [27]. Previous studies have also shown morphological adaptations of *L. gibbosus* to different habitats and competitors [28,29]. Currently, *L. gibbosus* is registered in at least 27 European countries and Turkey [23]. The European Union considers it among some of the most harmful invasive species and has included it in its list of "Invasive Alien Species of Union Concern", which obligates EU countries to take measures for its control [30,31].

Previous studies have shown that *L. gibbosus* is more adaptable to local conditions than other allochthonous species [12]. The species triggers changes in the food web of the streams it colonizes [18] and interacts trophically with native species, feeding on the eggs and juveniles of other species [32]. Although trophic ecology and ethology studies of *L. gibbosus*, which have informed the design and implementation of effective management tools [33], are relatively abundant, evolutionary studies are scarce. However, such studies are becoming increasingly important for understanding the mechanisms that make *L. gibbosus* such a successful invasive species [4,32–34]. For instance, recent evolutionary studies characterizing the genetic variability of *L. gibbosus* in its native area have demonstrated the existence of two differentiated populations, the Atlantic and the Mississippian [23], and that all the European populations studied thus far originate from the Atlantic population [23,35].

Prevention is a key factor in avoiding the successful establishment of invasive species, but it relies on understanding the origin and functioning of the processes involved [36]. Knowledge of how an invasive species first arrives, then establishes and spreads, especially under current climate change conditions, is critical for designing prevention and control measures. Despite the wide distribution of *L. gibbosus* and the serious consequences of its presence in the ecosystems it colonizes, there is still much to learn about the genetics and ecology of this species. The objectives of this study are to (1) analyze the genetic variability of the Mediterranean populations of *L. gibbosus* to identify their origin, dispersal routes, and relationship with native and other introduced fish populations in Europe, (2) evaluate which factors have favored its invasion of the Iberian basins, (3) investigate changes in climatic niche during the invasion process in Europe, (4) predict the future presence of the species worldwide, and (5) contribute to the development of management and control plans in areas that will foreseeably be invaded. We hypothesize that the European populations of *L. gibbosus* originated from a few introductions and, therefore, have low genetic variability, but after successfully establishing in Europe, these genetically depauperate populations have been able to expand their niche and, in turn, increase their invasive potential.

## 2. Materials and Methods

### 2.1. Sample Collection, DNA Extraction, Amplification, and Sequencing

Individuals of *L. gibbosus* were collected via non-selective electric fishing (HANS GRASSL EL62II motor) in nine basins of the Iberian Peninsula (Arade, Bañolas Lake, Ebro, Guadiana, Mira, Mondego, Sado, Tagus, and Vouga; see details in Table S1). A small portion of the pectoral fin of each individual was clipped and preserved in 95% ethanol for molecular analysis. Prior to genomic DNA extraction, the tissue was digested in 500 µL of Lysis Buffer (L15) and 10 µL of Proteinase K at 55 °C for 24 h. Genomic DNA was extracted using the Invitrogen ChargeSwitch® gDNA Tissue Kit, Carlsbad, CA, USA, following the manufacturer's instructions, and 100 µL of purified DNA was obtained for each sample.

Three mitochondrial genes were analyzed for a total of 56 individuals: cytochrome oxidase I (COI) (50 samples), the d-loop or control region (41 samples), and the ND1 subunit of NADH dehydrogenase (40 samples). The mitochondrial genes were amplified via polymerase chain reaction (PCR) using the conditions shown in Table 1. Each reaction contained 2.5 µL of 10× buffer, 2 µL of dNTPs (2.5 µM), 1.25 µL of primers (10 µM), 0.3 µL of Taq polymerase (1.5 U/µL; Takara, San Jose, CA, USA), 1–2 µL of DNA template and milliQ water to a final volume of 25 µL. After PCR verification, the samples were purified: for COI and d-loop, 2 µL of ExoSAP (10 µM; ThermoFisher, Lithuania) was used, and for ND1, the PCR bands were gel purified after the PCR product was evaporated to reduce the volume by half (final volume of 8 µL). Sequencing was performed by MACROGEN (Madrid, Spain) with the same primers used in the PCR.

**Table 1.** Gene-specific primer sequences and PCR conditions used in the study.

| | Primer | Sequence | PCR Conditions | | Reference |
|---|---|---|---|---|---|
| Cytochrome c oxidase subunit I (COI) | | | | 95 °C | 2′ | |
| | Fish-1F | 5′-TCAACCAACCACAAAGACATTGGCAC-3′ | | 95 °C | 30″ | [37] |
| | | | 35× | 54 °C | 30″ | |
| | Fish-1R | 5′-TAGACTTCTGGGTGGCCAAAGAATCA-3′ | | 72 °C | 1′ | [37] |
| | | | | 72 °C | 10′ | |
| | | | | 10 °C | ∞ | |
| Control region or D loop | | | | 94 °C | 2′ | |
| | CB3R-L | 5′-CAYATYMARCCMGAATGRTATTT-3′ | | 94 °C | 30″ | [38] |
| | | | 40× | 55 °C | 30″ | |
| | 12SAR-H | 5′-ATARTRGGGTATCTAATCCYAGTT-3′ | | 72 °C | 2′ | [39] |
| | | | | 72 °C | 5′ | |
| | | | | 10 °C | ∞ | |
| NADH dehydrogenase 1 (ND1) | | | | 95 °C | 5′ | |
| | L2949-LMA | 5′-AGTTACCCTAGGGATAACAGCGCAATC-3′ | | 95 °C | 45″ | [23] |
| | | | 10× (−0.5 °C) | 55 °C | 1′ | |
| | | | | 72 °C | 1′30″ | |
| | S2-LMA | 5′-GGTATGGGCCCAAAAGCTTA-3′ | | 95 °C | 45″ | |
| | | | 30× | 52 °C | 1′30″ | [40] |
| | | | | 72 °C | 1′30″ | |
| | | | | 72 °C | 6′ | |
| | | | | 10 °C | ∞ | |

### 2.2. Molecular Analysis

The sequences were visualized, aligned, and reviewed using Sequencher 5.0.1 (Gene Codes Corporation) and compared against a reference sequence for each marker available from GenBank (https://www.ncbi.nlm.nih.gov/genbank/, accessed on 28 October 2021).

Accession numbers are included in Table 2. Newly obtained *L. gibbosus* sequences from the Iberian Peninsula were analyzed with individuals from other introduced populations in 12 European countries (Austria, Belgium, Bulgaria, Croatia, France, Germany, Greece, Italy, The Netherlands, Norway, Slovakia, and the United Kingdom) and in Turkey, and from native populations in the USA and Canada. Available d-loop sequences were exclusively from native populations in the USA and Canada (see Table 2). Individual data matrices were constructed for COI (652 bp, 115 sequences), d-loop (849 bp, 46 sequences), and ND1 (975 bp, 101 sequences). Two combined matrices were also generated: one with the three genes analyzed (3321 bp, 32 sequences) and the other with only COI and ND1 (1588 bp, 37 sequences), given the scarcity of the d-loop data. A sequence of *Micropterus salmoides* was included in all matrices as the outgroup. All additional sequences included in this study were obtained from GenBank or, in the case of COI, the Barcoding Of Life Database (BOLD, https://www.boldsystems.org/, accessed on 28 October 2021) (Table 2).

**Table 2.** *Lepomis gibbosus* haplotypes obtained for COI (cH1–cH9), d-loop (dH1 and dH2), and ND1 (nH1–nH37). GenBank or BOLD (with *) identification numbers of *L. gibbosus* sequences used for the analysis. Individuals sharing a GenBank ID refer to the same haplotype extracted from Yavno et al. [23]. Country codes follow the nomenclature of Alpha-2.

| GenBank ID. | Locality | Country | COI | Reference | GenBank | Locality | Country | D-loop | Reference | GenBank | Locality | Country | ND1 | Reference |
|---|---|---|---|---|---|---|---|---|---|---|---|---|---|---|
| EU524723 | Quebec | CA | cH1 | [41] | MF621725 | Lake Erie | CA | dH1 | [42] | MN516524 | Quebec | CA | nH13 | [23] |
| EU524724 | Quebec | CA | cH1 | [41] | MF621724 | Pennsylvania | US | dH2 | [42] | MN516520 | Quebec | CA | nH14 | [23] |
| BCFB584-06 * | Quebec | CA | cH1 | [41] | MT667250 | Pennsylvania | US | dH1 | Unpublished | MN516505 | Quebec | CA | nH15 | [23] |
| BCFB585-06 * | Quebec | CA | cH1 | [41] | MF621726 | New York | US | dH1 | [42] | MN516501 | Quebec | CA | nH16 | [23] |
| BCFB586-06 * | Quebec | CA | cH1 | [41] | | Iberian Peninsula | ES/PT | dH1 | | MN516500 | Quebec | CA | nH17 | [23] |
| BCFB587-06 * | Quebec | CA | cH1 | [41] | | | | | | MN516496 | Quebec | CA | nH18 | [23] |
| BCFB588-06 * | Quebec | CA | cH1 | [41] | | | | | | MN516527 | Ontario | CA | nH6 | [23] |
| NHFEC079 * | Quebec | CA | cH1 | [41] | | | | | | MN516526 | Ontario | CA | nH7 | [23] |
| BCFB583-06 * | Quebec | CA | cH4 | [41] | | | | | | MN516525 | Ontario | CA | nH8 | [23] |
| EU524725 | Quebec | CA | cH4 | [41] | | | | | | MN516521 | Ontario | CA | nH9 | [23] |
| BCFB589-06 * | Ontario | CA | cH1 | [41] | | | | | | MN516523 | Ontario | CA | nH10 | [23] |
| BCFB590-06 * | Ontario | CA | cH1 | [41] | | | | | | MN516519 | Ontario | CA | nH11 | [23] |
| BCFB606-06 * | Ontario | CA | cH1 | [41] | | | | | | MN516499 | Ontario | CA | nH12 | [23] |
| BCFB592-06 * | Ontario | CA | cH2 | [41] | | | | | | MF621725 | Lake Erie | CA | nH37 | Unpublished |
| BCFB593-06 * | Ontario | CA | cH2 | [41] | | | | | | MN516514 | Wisconsin | US | nH28 | [23] |
| BCFB594-06 * | Ontario | CA | cH2 | [41] | | | | | | MN516509 | Wisconsin | US | nH29 | [23] |
| EU524717 | Ontario | CA | cH2 | [41] | | | | | | MN516508 | Wisconsin | US | nH30 | [23] |
| CFF194-16 * | Lake Erie | CA | cH1 | Unpublished | | | | | | MN516490 | North Carolina | US | nH34 | [23] |
| CFF048-16 * | Lake Erie | CA | cH5 | Unpublished | | | | | | MN516528 | Minnesota | US | nH2 | [23] |
| CFF104-16 * | Durand Lake | CA | cH1 | Unpublished | | | | | | MN516515 | Minnesota | US | nH3 | [23] |
| CFF107-16 * | Riviere Saint-Jean | CA | cH1 | Unpublished | | | | | | MN516507 | Minnesota | US | nH4 | [23] |
| CFF117-16 * | Opinicon Lake | CA | cH1 | Unpublished | | | | | | MN516513 | Minnesota | US | nH5 | [23] |
| CFF171-16 | Saint Louis Lake | CA | cH4 | Unpublished | | | | | | MN516502 | Pennsylvania | US | nH19 | [23] |
| RMAYB187-07 * | Ottawa | US | cH7 | [43] | | | | | | MN516498 | Pennsylvania | US | nH20 | [23] |
| SERCA041-12 * | Maryland | US | cH1 | [44] | | | | | | MN516493 | Pennsylvania | US | nH21 | [23] |
| UKFBJ845-08 * | New Hampshire | US | cH1 | Unpublished | | | | | | MF621726 | New York | US | nH34 | Unpublished |
| BNAFB548 * | Wisconsin | US | cH6 | Unpublished | | | | | | MN516497 | New Jersey | US | nH35 | [23] |
| HQ557271 | Wisconsin | US | cH6 | Unpublished | | | | | | MN516518 | Iowa | US | nH22 | [23] |
| RMAYB188-07 * | Wisconsin | US | cH8 | [43] | | | | | | MN516511 | Iowa | US | nH23 | [23] |
| BNAFB547-09 * | Wisconsin | US | cH6 | [43] | | | | | | MN516510 | Indiana | US | nH24 | [23] |
| JN026988 | South Carolina | US | cH1 | [43] | | | | | | MN516517 | Indiana | US | nH25 | [23] |
| RMAYB189-07 * | South Carolina | US | cH1 | [43] | | | | | | MN516516 | Michigan | US | nH26 | [23] |
| EPAMC397-20 * | Cincinnati | US | cH1 | Unpublished | | | | | | MN516512 | Michigan | US | nH27 | [23] |
| EPAMC398-20 * | Cincinnati | US | cH1 | Unpublished | | | | | | MN516522 | Virginia | US | nH31 | [23] |
| EPAMC399-20 * | Cincinnati | US | cH1 | Unpublished | | | | | | MN516503 | Virginia | US | nH32 | [23] |
| EPAMC400-20 * | Cincinnati | US | cH1 | Unpublished | | | | | | MN516504 | Virginia | US | nH32 | [23] |
| EPAMC401-20 * | Cincinnati | US | cH1 | Unpublished | | | | | | MN516494 | Virginia | US | nH33 | [23] |
| EPAMC402-20 * | Cincinnati | US | cH1 | Unpublished | | | | | | MN516495 | Virginia | US | nH33 | [23] |
| EPAMC403-20 * | Cincinnati | US | cH1 | Unpublished | | | | | | MN516492 | Virginia | US | nH34 | [23] |
| EPAMC404-20 * | Cincinnati | US | cH1 | Unpublished | | | | | | MN516489 | Virginia | US | nH34 | [23] |
| EPAMC405-20 * | Cincinnati | US | cH1 | Unpublished | | | | | | AB271766 | Hamilton | US | nH36 | [45] |
| EPAMC406-20 * | Cincinnati | US | cH1 | Unpublished | | | | | | | Iberian Peninsula | ES/PT | nH1 | |
| EPAMC407-20 * | Cincinnati | US | cH1 | Unpublished | | | | | | MN516506 | Monte Novo | PT | nH1 | [23] |
| EPAMC408-20 * | Cincinnati | US | cH1 | Unpublished | | | | | | MN516506 | River Ebro and | ES/PT | nH1 | [23] |
| EPAMC409-20 * | Cincinnati | US | cH1 | Unpublished | | | | | | MN516506 | Lake Bolsena | IT | nH1 | [23] |
| EPAMC410-20 * | Cincinnati | US | cH1 | Unpublished | | | | | | MN516489 | River Picocca | IT | nH34 | [23] |
| EFA105-16 * | Lake Superior | US | cH6 | Unpublished | | | | | | MN516506 | Argancy Pond | FR | nH1 | [23] |
| EFA207-17 * | Lake Superior | US | cH9 | Unpublished | | | | | | MN516489 | Birazel | FR | nH34 | [23] |

**Table 2.** *Cont.*

| GenBank ID. | Locality | Country | COI | Reference | GenBank | Locality | Country | D-loop | Reference | GenBank | Locality | Country | ND1 | Reference |
|---|---|---|---|---|---|---|---|---|---|---|---|---|---|---|
| | Iberian Peninsula | ES/PT | cH1 | | | | | | | MN516489 | Lake Prespa | GR | nH34 | [23] |
| HQ960772 | Hradec Králové | CZ | cH1 | Unpublished | | | | | | MN516506 | Mrzenica | RS | nH1 | [23] |
| ANGBF56447-19 * | Czech Republic | CZ | cH1 | | | | | | | MN516489 | Mrzenica | RS | nH34 | [23] |
| FBPIS137-10 * | Baviera | DE | cH1 | [44] | | | | | | MN516491 | Mrzenica | RS | nH34 | [23] |
| FBPIS140-10 * | Styria | AT | cH1 | [44] | | | | | | MN516489 | De Maten Ponds | BE | nH34 | [23] |
| FFMBH104-14 * | Turin | IT | cH1 | Unpublished | | | | | | MN516491 | Mastbos Pond | NL | nH34 | [23] |
| | | | | | | | | | | MN516489 | Mastbos Pond | NL | | [23] |
| FFMBH1592-14 * | Canton du Pilat | FR | cH1 | Unpublished | | | | | | MN516489 | Ogosta Reservoir | BG | nH34 | [23] |
| | | | | | | | | | | MN516489 | Slovakia | SK | | [23] |
| FFMBH2708-14 * | Sitagri | GR | cH1 | Unpublished | | | | | | MN516489 | Tanyard Fisheries | GB | nH34 | [23] |
| NOFIS136-18 * | Asker | NO | cH1 | Unpublished | | | | | | MN516489 | Sariçay Stream | TR | nH34 | [23] |
| JQ979159 | Ipsala | TR | cH2 | [37] | | | | | | | | | | |
| JQ979160 | Cayirkoy | TR | cH3 | [37] | | | | | | | | | | |
| JQ979161 | Bayraktar | TR | cH3 | [37] | | | | | | | | | | |
| JQ979162 | Davuldere | TR | cH3 | [37] | | | | | | | | | | |
| JQ979163 | Mugla | TR | cH4 | [37] | | | | | | | | | | |

*2.3. Phylogenetic Analysis*

To infer phylogenetic relationships, the two combined matrices were analyzed using Bayesian Inference (BI) and Maximum Likelihood (ML) approaches. For the BI analysis, we used MrBayes v3.0 [46], with 5 million generations of four simultaneous Markov chain Monte Carlo (MCMC) runs and a sample interval of 100 generations. We checked that the approximate standard deviation of the two parallel analyses, as a sign of convergence, was less than 0.01. Branch robustness of the obtained trees was checked with posterior probability (PP) values. Finally, after discarding the first 25% of trees of both analyses as burnin, we obtained the consensus tree, which was visualized and edited in FigTree v1.4.2 [47].

For the ML analysis, we used PhyML 3.0 on the ATGC platform [48]. First, the substitution model that best fits the data for the two combined matrices was selected under the Bayesian information criterion (BIC). This analysis showed that Hasegawa-Kishino-Yano 85 (HKY85), which assumes inequality between base frequencies and between transitions and transversions, best fit the data. Subsequently, we obtained the consensus tree of the two combined matrices, which were also visualized and edited in FigTree v1.4.2 [47]. The bootstrap (boot) method with 1000 replicates was used to assess the support and robustness of the branches.

*2.4. Analysis of Haplotypes, Genetic Distances, and Differentiation*

For the population analyses, we used DNASP 5.10.1 [49] and the individual matrices of the three genes studied. We calculated the haplotype and nucleotide diversity of the native and introduced populations of L. gibbosus. Haplotype networks were constructed for each gene, identifying individuals according to their geographical origin and analyzed in PopArt [50]. Finally, uncorrected genetic distances (p) between native and introduced populations of *L. gibbosus* were calculated for each gene in PAUP 4.0b10 [51] and clustered using Sequencer (program kindly provided by Bailey).

Genetic differentiation, as measured by the statistics $F_{ST}$, $F_{CT}$ and $F_{SC}$, was studied by an analysis of molecular variance (AMOVA) between the different geographical areas and for the three genes using Arlequin v3.5 [52].

*2.5. Occurrence Data and Climatic Variables*

Global occurrence data for *L. gibbosus* were downloaded from the Global Biodiversity Information Facility (GBIF, https://www.gbif.org/, accessed on 6 July 2022). In total, 83,249 records were reviewed, and those that lacked spatial information or were clearly erroneous (e.g., over the sea) were discarded. To eliminate possible collection bias, data points separated by a distance of less than 20 km were also removed using the package ecospat v3.2 [53] in R v.4.1.0 [54]. Based on the occurrence area, we generated two files, one corresponding to the native area, which included 4309 records from the basins of the St. Lawrence River and of other rivers in the eastern USA and Canada (including areas from which the predominant haplotypes in Europe originate [23]), and the second, to the invaded areas of Europe, which included 3376 records. In the second file, the occurrences were classified into three periods according to the date of introduction: (1) 1900–1959 based on data taken from De Groot [22], De-Sostoa et al. [55], and Elvira [25]; (2) 1960–1989, and (3) 1990–2021. Occurrences for the last two periods were based on the data available in GBIF.

Climate variables from WorldClim 2.1 [56], at a resolution of 2.5 min (~5 km at the equator), were used as explanatory variables. These variables have been shown to be key factors in regulating global fish distribution [57]. Variables showing irregular patterns in any of the study areas were removed from further analysis (Bio8, Bio9, Bio18 and Bio19). From each pair of variables with a Pearson correlation value greater than 0.8, we maintained that considered to have the greatest relative ecological importance for *L. gibbosus*, according to Manjarrés-Hernández et al. [58]. The final set included the variables Bio1 (annual mean

temperature, °C), Bio2 (mean diurnal range, °C), Bio4 (temperature seasonality, °C), Bio12 (annual precipitation, mm), and Bio15 (precipitation seasonality, mm).

### 2.6. Species Distribution Modeling (SDM)

Models were generated with MaxEnt v3.3.3k [59], which uses the principle of maximum entropy to estimate the probability of a species' potential distribution using known presences together with a set of predictor variables. MaxEnt only works with presences and generates the environmental comparison space from an area geographically accessible to the modeled species, or 'background' in MaxEnt terminology. These background points can be generated automatically by the software or inputted manually by the researcher. Background points in this study were generated within the watersheds inhabited by the species using information available at https://www.sciencebase.gov/catalog/item/4fb697b2e4b03ad19d64b47f, accessed on 3 August 2022 for America, and https://www.eea.europa.eu/data-and-maps/data/european-river-catchments-1/zipped-shapefile-vector-polygon/zipped-shapefile-vector-polygon/at_download/, accessed on 3 August 2022 for Europe.

We first ran several exploratory models to determine the regularization value that produced smooth response curves without inflating the number of parameters and found that a regularization value of $\beta = 1$ generated accurate models without overfitting. Then, we ran three models that were generated with other parameters set to default ('Auto features' and maximum number of iterations = 500). The first model (native) consisted of the native presence and background data; the second (introduced), the presence and background data from the introduced areas; and the third (mixed), the presence and background data of both native and introduced areas. All models were projected at a global scale and under current climatic conditions. The obtained models were also evaluated using the AUC and projected to two future climate periods (immediate 2041–2060 and distant 2081–2100) at a global scale and under two Representative Concentration Pathways (RCPs), one 'optimistic' (RCP4.5), in which emissions peak around 2040 and then decrease, and one 'pessimistic' (RCP8.5), in which emissions will continue to increase throughout the century. Data were gathered from WorldClim 2.1 at a spatial resolution of 10 min (~20 km at the equator), using the theoretical general circulation model (GCM) CNRM-ESM2-1.

### 2.7. Niche Overlap Analyses

To test if climatic niche differs between pairs of populations, we used the package 'phyloclim' 0.9.5 [60] in R following the methodology of Rodríguez-Merino et al. [5]. We calculated Schoener's *D* and Hellinger's *I* distance to analyze niche overlap between pairs of populations in order to determine whether the observed niches are more similar than would be expected. We also ran a background similarity test to analyze whether observed niche overlap can be attributed to differential use of the available environmental space by different populations. The former is used for sympatric distributions, while the latter is used for allopatric populations [61,62]. The identity hypothesis is rejected when the observed value of the similarity indices (*D* or *I*) is significantly lower or higher than those expected at random. The values of both measures lie between 0 (which indicates that niches do not overlap) and 1 (complete overlap). A randomization of 100 replicates was performed to generate the null distribution. Pairwise comparisons were made between the native and the 1900–1959, 1960–1989, and 1990–2021 introduction models for a total of six comparisons for each test.

## 3. Results

### 3.1. Molecular Characterization and Phylogenetic Analysis

Sequences of COI were successfully obtained from 50 individuals [613 base pair (bp) alignment with 16 variable and 11 parsimony–informative sites]. For the d-loop, 849 bp were amplified from 41 individuals (one parsimony–uninformative, variable site), and for ND1, 975 bp from 40 individuals (42 variable and 23 parsimony-informative sites).

Although none of the mitochondrial genes analyzed could be considered highly variable, ND1 was the most variable of the three. No significant differences in nucleotide frequencies between individuals were found for any of the markers (COI: $X^2 = 0.37$, *p* = 1.0; d-loop: $X^2 = 0.16$, *p* = 1.0; ND1: $X^2 = 5.18$, *p* = 1.0).

The phylogeny reconstructions obtained by BI and ML approaches with the two combined matrices (COI + ND1 or the three genes) were congruent (Figures 1 and S1). In the phylogeny obtained with the combined matrix of the three genes (Figure S1), the Iberian individuals grouped together, forming a moderately supported monophyletic group (PP = 0.80 and boot = 78 for BI and ML, respectively). This group resolved as related to the group comprising the US populations (PP = 0.98, boot = 82), with the Canadian populations being the most distantly related group.

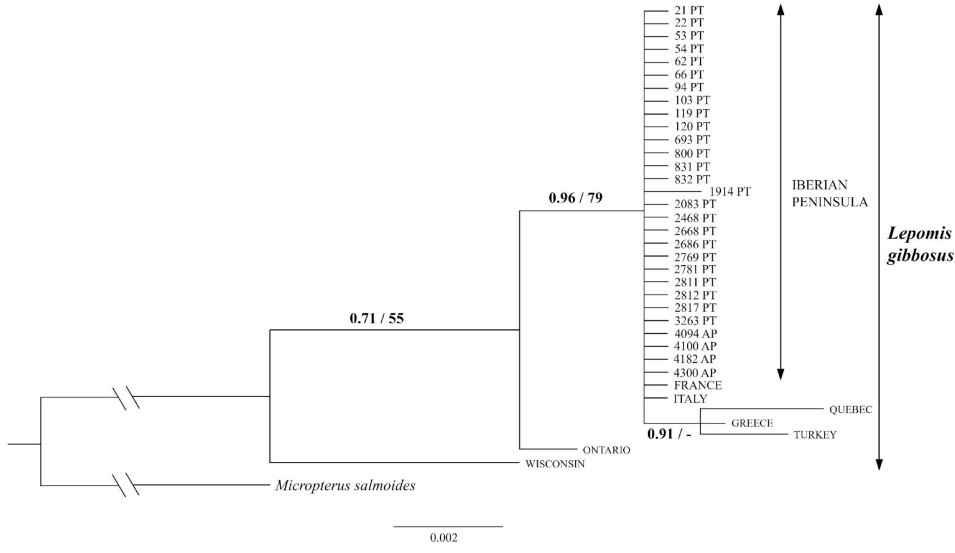

**Figure 1.** ML phylogenetic tree of *Lepomis gibbosus* from native and introduced populations based on the combined matrix of two genes (COI and ND1). Values on branches correspond to the posterior probability (PP) and bootstrap (boot) values—indicates no support. Label identification refers to ID numbers in Table S1.

Greater structuring is observed in the phylogeny based on the combined two-gene matrix, which, in addition to the native Canadian and US populations, includes those introduced to European and Turkish waters (Figure 1). The Iberian populations, together with individuals from France and Italy, form a monophyletic group. This group is closely related to some individuals native to Canada (Quebec) and to those introduced to Greece and Turkey (PP = 0.96, boot = 79), followed by the group composed of the rest of the individuals from the native populations of Canada (Ontario) (PP = 0.71, boot = 55). The US population appears to be the most divergent group (PP = 1, boot = 100).

### 3.2. Population Analysis: Haplotype Network, Genetic Distances, and Differentiation

Haplotype networks were constructed for each gene (Figure 2). Native populations presented the highest number of haplotypes (Table 2). For COI, nine haplotypes (cH1–cH9) were found (115 individuals). Eight of these (all except cH3) were found in the native populations, and cH1 was the most common (found in 10 of the 14 populations analyzed) (Figure 2a). For d-loop, the two haplotypes found (dH1 and dH2; 46 individuals) were present in both native and introduced populations (Figure 2b). The highest number of haplotypes, 37, was found for the ND1 gene (nH1–nH37; 81 individuals). Of these, 36 were present in the native populations (nH2–nH37) (Figure 2c), and the most common haplotype (nH2) was present in almost all the introduced populations (except the analyzed Iberian populations that presented the nH1 haplotype) and some US populations.

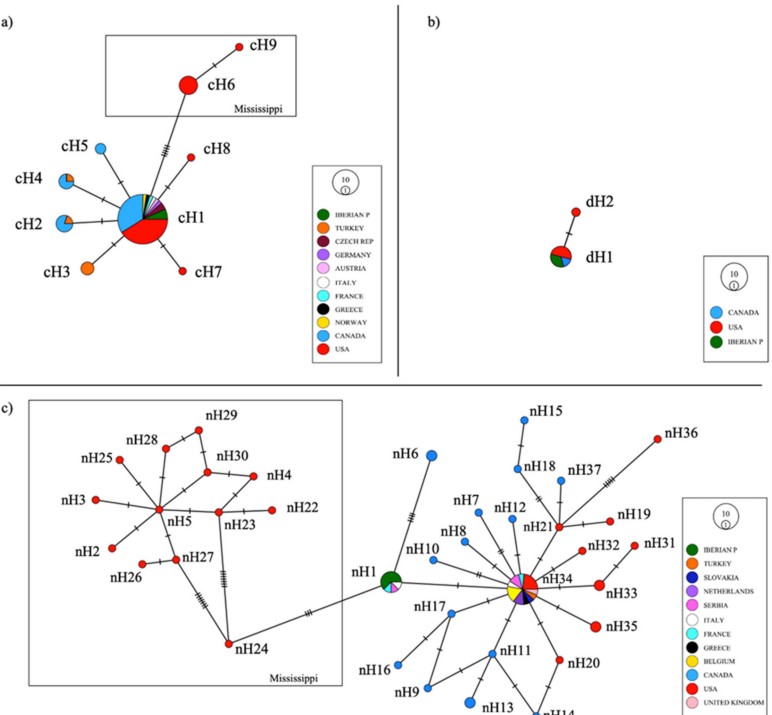

**Figure 2.** Haplotype networks of *Lepomis gibbosus* from native and introduced populations. (**a**) COI, (**b**) d-loop, and (**c**) ND1. Circle size is proportional to the number of individuals sharing the haplotype. Each haplotype is identified with a numerical code (as in Table 2), and its geographical origin is indicated by different colors. Mutation steps between haplotypes are represented by a cross-hatch line.

The introduced populations in Europe and Turkey have a small number of haplotypes for all genes. For COI, only cH1 is present in the European populations; in Turkey, three haplotypes (cH2–cH4) are found, with one being unique to this area (cH3). For ND1, the European populations present two interesting haplotypes, nH1 and nH34: nH1 is exclusive to some European populations, including the Iberian ones, and nH34 is shared between Europe, Turkey (where it is the only ND1 haplotype), and some native populations from North America. Interestingly, all individuals from the Iberian populations shared the same haplotype for each of the genes (i.e., cH1, nH1, and dH1). In the case of the COI and d-loop haplotypes, cH1 and dH1 are also shared with the native populations; by contrast, the ND1 haplotype nH1 is shared with only some of the European populations (France, Serbia and Italy) and is not found in the native populations.

The highest values of haplotype and nucleotide diversity were found for ND1 (hd = 0.776 ± 0.00159, sd = 0.040, and nd = 0. 00402), followed by COI (hd = 0.356 ± 0.00326, sd = 0.057, and nd = 0.00168), and finally d-loop (hd = 0.048 ± 0.002, sd = 0.045, and nd = 0.00006). With respect to native versus introduced populations, higher values of nucleotide and haplotype diversity were observed in the native (ND1 = 0.00757, COI = 0.00323, and d-loop = 0.00050) versus introduced ones (ND1 = 0.00036, COI = 0.00030, and d-loop = 0.0).

The largest genetic distances (uncorrected p) were observed between the native populations (USA and Canada) for each of the genes (0.32 COI, 0.76 ND1, and 0.05 d-loop) (Table 3). In the case of ND1, the Mississippi and Atlantic native populations presented the largest genetic distance observed (1.30), followed by the distance between the Mississippi and European populations (1.15). Genetic distances between native and European populations were smaller than between native populations: a distance of 0.25 and 0.49 was observed for COI and ND1, respectively (genetic distances between European populations could not be calculated for d-loop due to a lack of data). The genetic distances between Iberian and native populations were similar to the ones observed in comparison with all the European populations: 0.18 for COI, 0.52 for ND1, and 0.02 for d-loop.

**Table 3.** Uncorrected (p) mean distances based on the individual gene matrices for *Lepomis gibbosus*. Values are means ± standard deviation {range}. Gene Mean refers to mean values for each gene: Within = Inside populations; Between = Between populations.

| Gene | Group | America | Europe | Iberian Peninsula | Gene Mean | | |
|------|-------|---------|--------|-------------------|-----------|---|---|
| COI | America | 0.32 ± 0.42 {0.0–1.36} | | | COI | 0.16 ± 0.33 {0.00–1.36} | Within |
| | Europe | 0.25 ± 000.35 {0.00–1.35} | 0.13 ± 0.12 {0.00–0.39} | | | 0.17 ± 0.32 {0.00–1.35} | Between |
| | Iberian P | 0.18 ± 0.34 {0.00–1.16} | 0.07 ± 0.09 {0.00–0.19} | 0.00 ± 0.00 {0.00–0.00} | | | |
| ND-1 | America | 0.76 ± 0.53 {0.00–1.95} | | | ND-1 | 0.36 ± 0.53 {0.00–1.95} | Within |
| | Europe | 0.49 ± 0.47 {0.00–1.33} | 0.03 ± 0.05 {0.00–0.10} | | | 0.42 ± 0.42 {0.00–1.33} | Between |
| | Iberian P | 0.52 ± 0.42 {0.10–1.33} | 0.08 ± 0.04 {0.00–0.10} | 0.00 ± 0.00 {0.00–0.00} | | | |
| D-loop | America | 0.05 ± 0.06 {0.00–0.12} | | | D-loop | 0.00 ± 0.01 {0.00–0.12} | Within |
| | Iberian P | 0.02 ± 0.05 {0.00–0.12} | 0.00 ± 0.00 {0.00–0.00} | | | 0.02 ± 0.05 {0.00–0.12} | Between |

With respect to the genetic differentiation of populations, our analyses indicate that the highest level of genetic differentiation is between the introduced and native Mississippi populations. The values of genetic differentiation were particularly supported by the analysis of ND1, for which it was possible to analyze Atlantic, Mississippi, and introduced populations from both Europe and Turkey. Much lower values of genetic differentiation were found between the introduced and Atlantic populations. Finally, genetic differentiation within the introduced populations was very low, with values of near 0.

### 3.3. Species Distribution Modeling

The AUC values were all greater than 0.76 (native AUC = 0.780, introduced AUC = 0.783, and mixed AUC = 0.761). Despite being considered low, such values are often observed for generalist species. The native model projections predict the Mississippi basin, where the other native group of *L. gibbosus* is currently found, as a high suitability area (Figure 3). Very low suitability was predicted for the rest of the world, except very specific areas of the Po River basin in the northern Italian Peninsula. The introduced model projections showed suitability in a greater number of areas and more globally distributed: in addition to the native area, suitable areas include the southern tributaries of the Mississippi basin in North America, the upper Amazon basin in Peru and Ecuador, large areas around Rio de la Plata and the coastal border of Brazil (Bahia State) in South America, across almost all of Europe, and large areas of western Australia, Borneo, Sulawesi, and New Guinea (Figure 3). The mixed model predicted high suitability in Europe and in the Rio de la Plata area in South America, similar to the introduced model, but lower suitability in the native areas of North America than the native model projections (Figure 3). Areas of Australia, Borneo, Sulawesi, and New Guinea are also suitable but with low values.

The models (native, introduced, and mixed) were also analyzed with consideration of future climate scenarios. The projected species distribution of *L. gibbosus* under the future scenarios is largely the same as described above for present-day climate conditions, except that suitability increases in the core areas and expands in a regular form in accordance with the present climate (Figures 3 and 4).

### 3.4. Niche Identity and Overlap Analysis

According to the results of the niche identity tests, the niche generated with presences from 1900 to 1959 does not differ from that found from 1960 to 1989 ($D = 0.671$, $I = 0.87$) or 1990 to 2021 ($D = 0.597$, $I = 0.851$), which would be the consequence that occurrences in the 1900 to 1959 dataset are included within the other two (Figure 5). Interestingly, even though the 1960–1989 presences are a geographically more limited subset of the 1990–2021 ones, niche identity could be rejected.

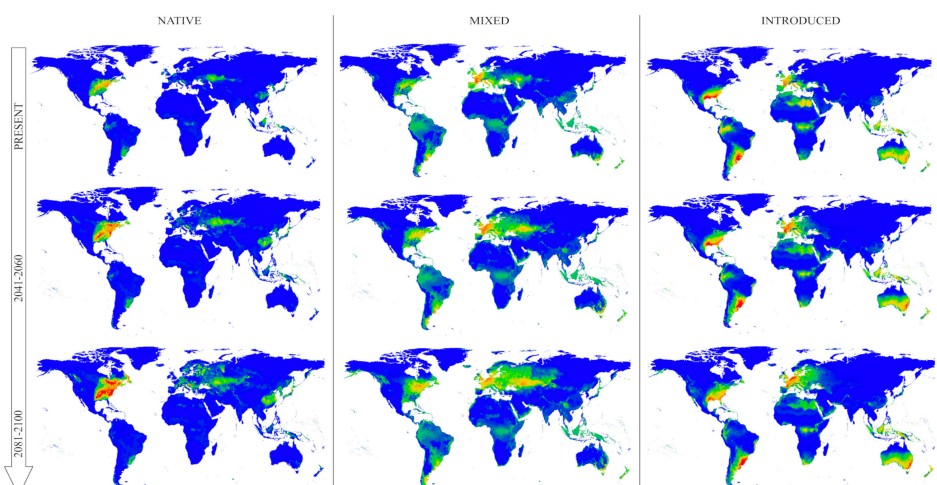

**Figure 3.** Potential distribution models of *Lepomis gibbosus* obtained with MaxEnt for the present, immediate future (2041–2060), and distant future (2081–2100) under a pessimistic emissions scenario (RCP8.5). Suitability values are represented by a color scale: areas of non-suitability in blue, medium suitability in green-yellow and suitable in red.

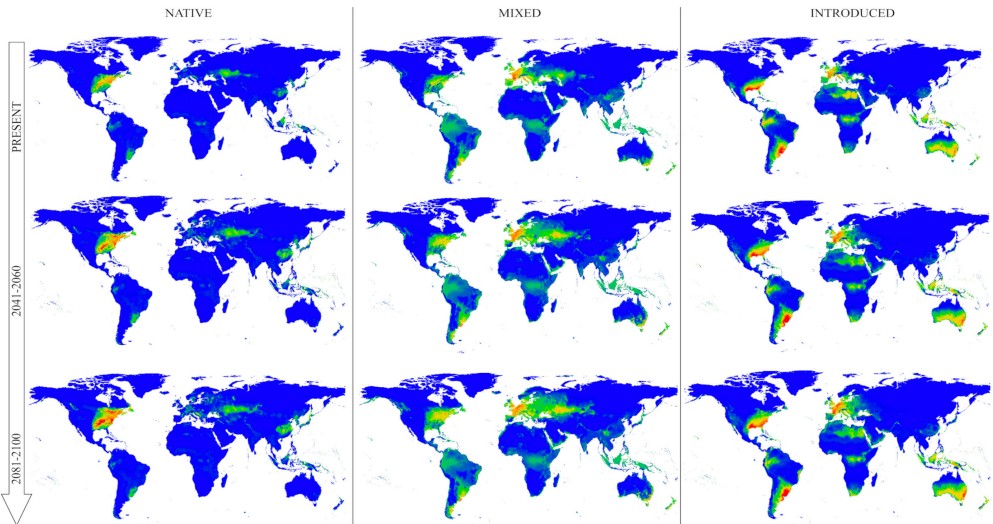

**Figure 4.** Potential distribution models of *Lepomis gibbosus* obtained with MaxEnt for the present, immediate future (2041–2060), and distant future (20812100) under an optimistic emissions scenario (RCP4.5). Suitability values are represented by a color scale: areas of non-suitability in blue, medium suitability in green-yellow and suitable in red.

In the case of the background similarity test between native and introduced populations in Europe, the result depends on the direction of the comparison. When the native model was compared with random ones generated using the background of the introduced area, the niches of the two populations were significantly more alike than different, despite the great environmental differences between the two areas. However, when the introduced model (generated using European occurrences) was compared with random ones generated using the native background, the niches differed more than would be expected under the null hypothesis, which assumes niche differences are exclusively due to differences in available habitat.

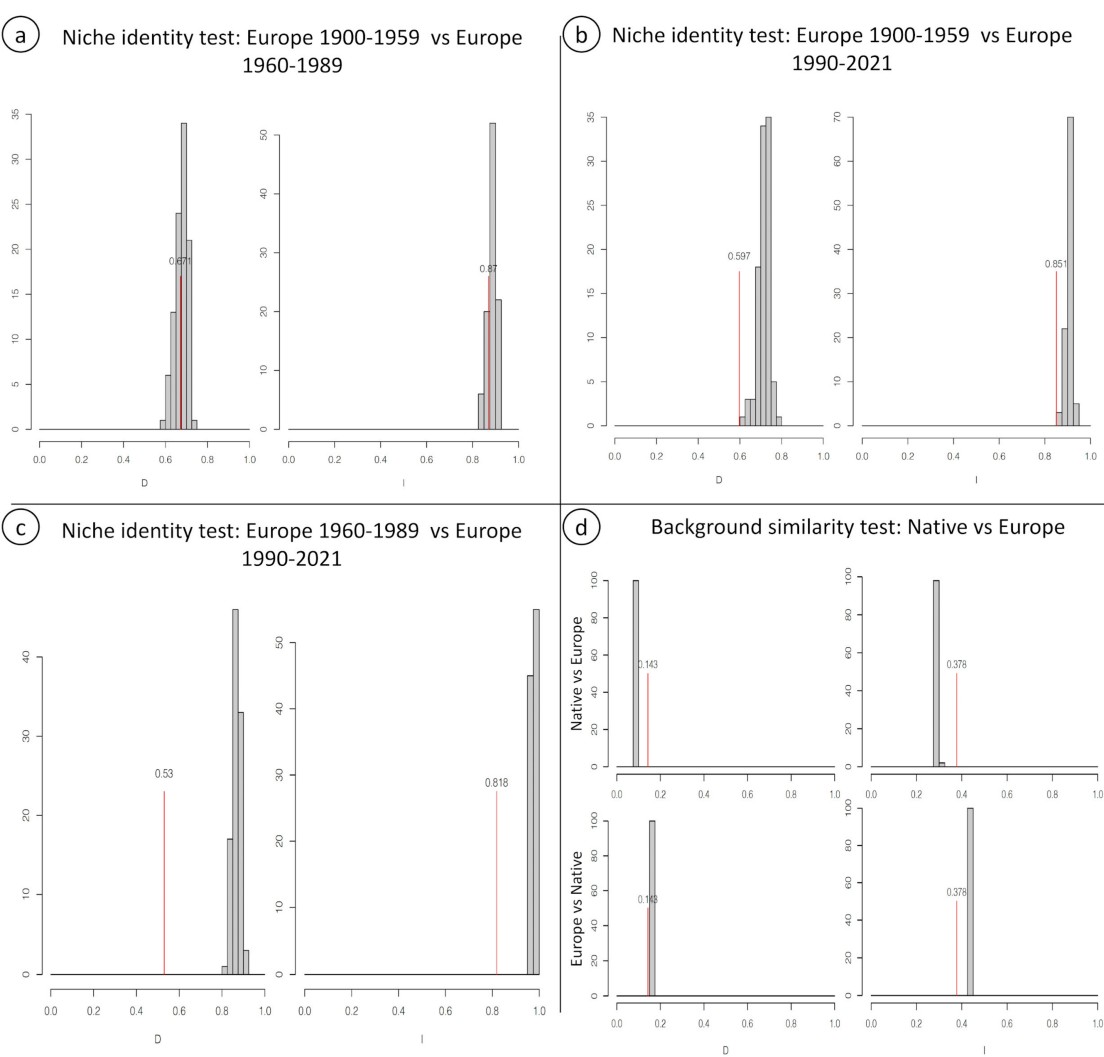

**Figure 5.** Plots obtained from the niche identity (**a–c**) and habitat similarity (**d**) tests. (**a–c**) Plots showing the identity tests, using Schoener's D (D) and Hellinger's I distance (I), between the introduced niches of Europe at different times. The red line indicates the identity between niches. The bars indicate the randomly generated null distribution. (**d**) Graphs showing the similarity test, also using Schoener's D (D) and Hellinger's I-distance (I), between introduced and native habitats. The red line indicates the similarity between habitats. The bars show the randomly generated null distribution.

## 4. Discussion

The results of our study support the idea that the European populations of *L. gibbosus* were established from a small number of introductions from its native distribution in eastern North America at the end of the 19th century. These introduced populations then acted as sources for further spread of the species to other European areas, including the Iberian Peninsula and Turkey. We also show that, during the ~140-year process of introduction of the species to mainly cool European areas, the species' niche has changed. Some populations in Europe are now able to withstand much warmer conditions than their native source populations, i.e., Iberian and, thus, have a high potential to invade even warmer areas. This potential increases the threat that *L. gibbosus* already has on the freshwater systems they inhabit. Considering the temperature increase projected due to global warming, this threat will only increase.

### 4.1. Origin and Dispersal of Lepomis gibbosus in Europe, Turkey, and the Iberian Peninsula

Two distinct native populations of *L. gibbosus* (Mississippian and Atlantic) are currently known. These populations, along with the populations that have been introduced to at least

28 countries since the late 19th century [32], diverged during the late Pleistocene [63]. Our results, based on the analyses of three mitochondrial genes, confirm a clear reduction in genetic variability in the invasive populations, likely due to a founder effect and subsequent bottleneck [64]. This finding suggests that the introduced populations in Europe originated from a small number of individuals from a few localities of the Atlantic native population, supporting the conclusions of Yavno et al. [23]. Our finding that all the Iberian populations and some from France, Italy, and Serbia lack the nH1 haplotype also supports the origin of these populations from a small source population pool (or suggests this haplotype is present in a low proportion of the studied populations and has not yet been sampled). This result could be a consequence of a founder effect followed by a bottleneck and genetic drift that occurred after a small number of native individuals from populations with a low proportion of nH1 were introduced to Europe. Based on the shared haplotypes and the low genetic distances observed, the Iberian populations most likely originated from the French stock. Indeed, many invasive species are thought to have been introduced to southern and northern Europe via France, which is considered both a "recipient" and "donor" country [11,65]. Colonization from France to Spain and then Portugal via shared rivers has been demonstrated in other freshwater fish species, such as the roach (*Rutilus rutilus*) and the bleak (*Alburnus alburnus*) [66]. According to our data, this route most likely drove the invasion of *L. gibbosus* in European rivers.

With respect to the Turkish populations, a previous study has suggested that they are related to adjacent Greek river populations based on their shared genetic diversity and some ecological factors [33]. However, in our study, the Greek and Turkish populations do not share any haplotypes. According to our COI haplotype network, the Turkish populations are related to native populations in Quebec and Ontario (Canada), with which they share the cH2 and cH4 haplotypes.

*4.2. Niche Shift and Ecological Success*

Although native and invasive populations of *L. gibbosus* do not present a significant difference in dispersal tendency [67], possibly owing to the decline over time in the adaptive value of rapid dispersal phenotypes, the European populations significantly increased their niche breadth during the invasion process (see Figure 5). Our niche identity tests revealed no significant differences in niche similarity between the 1900–1959 and 1960–1989 invasion periods; however, a remarkable decrease in similarity was observed between the 1960–1989 and 1990–2021 periods. From 1960 to 1989, large permanent populations were established in the rivers of central European countries with cool climates and stable water regimes following the successful introduction of native individuals from the Atlantic population that inhabited a similar climatic area. However, from 1991 to 2021, translocations from those central European populations were successfully established in warmer areas of southern and eastern Europe that have a more seasonal water regime. Overall, these findings further support the high adaptability of *L. gibbosus* shown in previous studies [24,68].

The habitat similarity tests comparing native versus introduced habitats against different backgrounds show that the niches of the native and introduced populations were more similar than expected when the native populations were compared against the introduced background but less similar when the introduced European populations were compared against the native background. These results confirm that the invasive populations experienced a niche shift toward warmer waters and more intermittent water regimes driven by high precipitation seasonality. Our results are in line with those of Rooke and Fox [69], who concluded that current European populations of *L. gibbosus* could not withstand the low temperature ranges typical of the native range. Given the predicted rise in global temperatures, the evidence suggests that the invasive potential of the introduced populations of *L. gibbosus* will only increase, and so will their threat.

Our habitat suitability projections for *L. gibbosus* show that the potential future range of this sunfish will increase worldwide, particularly considering the multiplicative invasive potential of its native populations plus that of the older (cooler) and the more recently es-

tablished (warmer) European populations. Our projections agree with those of Artaev [70], who concluded that climate change will cause *L. gibbosus* to spread to new suitable areas in northern and eastern Europe for the foreseeable future, as these areas will represent a moderate change compared with their existing areas. Therefore, this species can be regarded as a potentially dangerous invasive species throughout most of Europe (and likely beyond). Many areas currently free of this pest could be invaded in the near future if measures are not taken to control existing populations and to prevent their spread. In fact, researchers expect that this sunfish will soon colonize the African continent, where other species that normally coexist with it, e.g., *Micropterus salmoides*, are already present [11]. This hypothesis is also supported by our future projections. The high invasive potential and projected future distribution of *L. gibbosus*, which can adapt and establish in a wide range of environments by rapidly shifting its niche [71,72], should cause great concern. For example, without management measures, most cool-water rivers of Western and Southern Europe could be invaded by 2060. The extent of invasion, however, would be modulated by the morphostructure of the different river basins: short rivers with a high elevation gradient and high dissolved oxygen concentrations, for instance, would be less affected.

Despite the low genetic diversity of the species, *L. gibbosus* exhibits a high level of morphological and ecological plasticity [34,73]. This plasticity, combined with the species' ability to adapt and expand niches, makes the pumpkinseed an ideal invader, in agreement with previous invasive species risk assessments [74]. New prevention measures that consider the biology, ecology, and genetics of *L. gibbosus* are necessary and would require coordinated legislation among European countries, transnational management plans, and early warning measures. The role of climate change (increased temperatures and reduced water flow) must also be taken into account in these plans, as it will influence the potential establishment of the species in currently uninhabitable areas whose habitat suitability will likely increase in the future.

**Supplementary Materials:** The following supporting information can be downloaded at: https://www.mdpi.com/article/10.3390/d15101059/s1. Figure S1: ML phylogenetic tree of Iberian and native populations from USA and Canada of *Lepomis gibbosus* based on the combined three-gene matrix (COI, d-loop, and ND1). Values on branches correspond to posterior probability (PP) and bootstrap (boot) values; Table S1: Detailed information on the Iberian populations of *Lepomis gibbosus* analyzed in the present study. GenBank accession numbers will be added upon publication.

**Author Contributions:** Conceptualization, F.M., J.M. and A.P.; methodology, A.L.-C., F.M., J.M. and A.P.; software, A.L.-C. and J.M.; validation, F.M., J.M. and A.P.; formal analysis, A.L.-C., F.M., J.M. and A.P.; investigation, A.L.-C., F.M., J.M. and A.P.; resources, F.M., J.M. and A.P.; writing—original draft preparation, A.L.-C.; writing—review and editing, A.P., F.M. and J.M.; visualization, A.L.-C., F.M., J.M. and A.P.; supervision, F.M., J.M. and A.P.; project administration, F.M., J.M. and A.P.; funding acquisition, A.P. All authors have read and agreed to the published version of the manuscript.

**Funding:** This research received no external funding.

**Institutional Review Board Statement:** Not applicable.

**Data Availability Statement:** Sequences of this study are available at https://www.ncbi.nlm.nih.gov/.

**Acknowledgments:** We are grateful to A.R. Amaral, M.M. Coelho, C. Cunha, N. Franch, L. Méndez, Q. Pou, F. Ribeiro, and G. Sousa for help with specimen collection in the field. We thank M. Gutiérrez Ray for her help in the lab. Thanks to M. Modrell, who conscientiously revised the language. This work has corresponded to the master's thesis of A. Lambea in the master of Biodiversity in tropical areas and its conservation (CSIC-UIMP).

**Conflicts of Interest:** The authors declare no conflict of interest.

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
