# Peer review of "Genetic and Ecological Approaches to Introduced Populations of Pumpkinseed Sunfish (Lepomis gibbosus) in Southwestern Europe"

_diversity, doi:10.3390/d15101059_

Round 1

Reviewer 1 Report

The article “diversity-2567840-Genetic and ecological approaches to introduced populations of pumpkinseed sunfish (/Lepomis gibbosus/) in southwestern Europe “aims to contribute to the understanding of the potential impacts of the exotic /L. gibbosus/ introduced in continental waters of southwestern Europe. The information presented here allows us to observe the tendency of this introduced species to increase its impacts on habitats and fish populations and native biodiversity in southwestern Europe. The genetic data and its application to environmental information support the event of a potential increase in the dispersion of this introduced exotic fish. In my opinion, the article is relevant to this area of science, providing a set of data in support of the advancement of these topics.

The article is clear, written in an appropriate English way and relevant for the field and presented in a well-structured manner, but I strongly suggest the use of the Microsoft Word Template included in the Instructions for Authors of the Journal page. Cited references are relevant and some are recent (within the last 5 years 6% y 10 years 36%). There is not lack of citations in the bibliography section and there is not an excessive number of self-citations.

The study is correctly designed and sufficiently sound with enough information to draw a conclusion. Methods are described with sufficient details to allow another researcher to reproduce the results The data and analyses presented is appropriately and agree with the conventional methods and results outputs and results reproducible based on the details given in the methods.

Some suggestions are included in the attached article file using Word tracking changes related to the proper recognition of labels used in phylogenetic trees as well as the presentation of table information.

Since there are not many articles published with information that supports the management of introduced species from the genetic and global environmental point of view, this article is novel and is a clear contribution to the knowledge of this topic.

The article fits in the subject area of Organismic diversity and diversity preservation, including analysis Genetic diversity and phylogeny for the potential impact of Invasive organisms in a global change effect scenario.

Although a section was not written for the conclusion of the article, the discussion definitely directs the contribution of the article to a conclusion about the approaches raised.

I believe that the availability of this and other articles on this subject will attract more scientific readers, since it is an area of generation of the necessary knowledge for the management of biodiversity in the face of global climate change. 

Author Response

Suggestions of Reviewer 1 are indicated in blue in the attached manuscript sent to Editor. L18. Ten has been changed by "10". L111. We consider "each individual" more correct than "everyone", and thus we have maintained it. L128. Table S1 heading. We have added Country codes, ID and H1 definitions. We considered lat & long coordinates very informative columns, therefore we have maintained then. L130. He have maintained the name the of section "Molecular analysis". L256. We found more adequate to maintain "BI and ML" for all readers to understand. L268. Figure 1 caption. We have specified the correspondence between specimen names and ID numbers in Table S1. L388. We have specified the populations. L394. Italics removed. L450. Changed: "are in agreement with" by "agree with" L490. "Bibliography" has been changed by "References"

Reviewer 2 Report

Interesting and valuable research on one of the significant invasive species. I've almost no concerns, only a few small notes - links on the lines 210-211 are not opening properly, and figures 3-4 are too small and of bad quality, it's really hard to detect differences between different parts of figure.

Author Response

Suggestions of Reviewer 2 are indicated in yellow in the attached manuscript sent to the Editor.. L210-211. We have modified the links and now they are working. L350. Figures 3 and 4. We have increased the size of the maps to the maximum in order to keep all maps in the same figure. If with this increase the maps are not well visualized, we could send the 18 maps separately including them as supplementary material.

Reviewer 3 Report

I am inclined to regard this manuscript as of high quality, made on sufficient material and with the use of adequate methods of analysis. I also find it interesting that the authors combined two approaches in one study, although each of them could be the material for a separate article. Studying another invasive fish species in another region, I see great parallels in the results. I will wait for the publication of this manuscript so that I can cite it in our report. I think the article will be of interest to a wide range of readers.

The originality of the text of the manuscript is more than 85%. Although I am not a native speaker, the English was clear and easy to read.

Minor typos and errors in the bibliographic list have been noticed. For example, there are typos on lines 467,494, 520, 551, 571, 586, 609, 611, 766. Some of the attached drawings (pages 40, 41) are not of very good quality.

The manuscript can be accepted for publication after minor technical editing.

Author Response

Suggestions of Reviewer 3 are indicated in orange in the attached manuscript sent to the Editor.. The typos on lines 467, 494, 520, 551, 571, 586, 609, 611, and 766 are corrected and highlighted in orange. We send new Figures 1-5 and Figure S1 and 2.